# Porcine Reproductive and Respiratory Syndrome (PRRS) and *CD163* Resistance Polymorphic Markers: What Is the Scenario in Naturally Infected Pig Livestock in Central Italy?

**DOI:** 10.3390/ani13152477

**Published:** 2023-07-31

**Authors:** Martina Torricelli, Anna Fratto, Marcella Ciullo, Carla Sebastiani, Chiara Arcangeli, Andrea Felici, Samira Giovannini, Francesca Maria Sarti, Marco Sensi, Massimo Biagetti

**Affiliations:** 1Istituto Zooprofilattico Sperimentale dell’Umbria e delle Marche-Togo Rosati (IZSUM), Via Salvemini 1, 06126 Perugia, Italy; m.torricelli@izsum.it (M.T.); a.fratto@izsum.it (A.F.); m.ciullo@izsum.it (M.C.); a.felici@izsum.it (A.F.); m.biagetti@izsum.it (M.B.); 23A-Parco Tecnologico Agroalimentare dell’Umbria, 06059 Todi, Italy; carcangeli@parco3a.org; 3Dipartimento di Scienze Agrarie, Alimentari e Ambientali, University of Perugia, 06121 Perugia, Italy; samira.giovannini@gmail.com (S.G.); francesca.sarti@unipg.it (F.M.S.); 4Independent Researcher, 06083 Perugia, Italy; marsenvet@gmail.com

**Keywords:** PRRS, naturally exposed and/or infected pigs, genetic markers, *CD163*, polymorphisms, SNP, Italian pig livestock

## Abstract

**Simple Summary:**

Porcine Reproductive and Respiratory Syndrome (PRRS) is an infectious disease of viral etiology characteristic of the swine species. Although almost three decades have passed since its emergence, control of the disease still presents significant issues, representing a source of concern for veterinarians and breeders. In addition to management difficulties, the PRRS virus (PRRSV) causes severe economic losses in terms of abortions, a decrease in animal growth, increased mortality, and the massive use of drugs. Understanding the genetic markers involved in the response to the infection is challenging and crucial and represents the main goal of this study. *CD163*, which encodes the membrane receptor used by the PRRSV to enter macrophages and initiate infection, has been identified as one of the most promising marker genes associated with genetic susceptibility to the disease. In this study, detection by sequencing of the more significant polymorphisms on the *CD163* gene was conducted for the first time on 377 pigs reared in different farms distributed in some areas of Central Italy. The genotyping data obtained in this work, together with the assessment of the virological status of the animals and the comparison with the findings from other PRRSV conditioned and experimental infection trials, will allow a better understanding of whether some Italian pig populations can represent a good genetic resource and a reservoir of resistance/“resilience” markers to PRRS. Indeed, marker-assisted selection (MAS) could represent an alternative and a more valid tool than vaccination to control the spread of this impactful disease.

**Abstract:**

Porcine Reproductive and Respiratory Syndrome (PRRS) caused by the PRRS virus affects farmed pigs worldwide, causing direct and indirect losses. The most severe manifestations of PRRS infection are observed in piglets and pregnant sows. The clinical outcome of the infection depends on the PRRSV strain’s virulence, the pregnancy state of the female, environmental factors, the presence of protective antibodies due to previous infections, and the host’s genetic susceptibility. The latter aspect was investigated in this study, in particular, evaluating the most significant polymorphisms (SNPs) of the *CD163* gene in slaughtered pigs reared in Central Italy. Total RNAs were extracted from 377 swine samples and subjected to RT-PCR targeted to the *CD163* gene, followed by sequencing analysis. Contextually, the viral RNA was detected by RT-qPCR in order to phenotypically categorize animals into infected and not infected. In particular, 36 haplotypes were found, and their frequencies ranged from 0.13% to 35.15%. There were 62 resulting genotypes, three of which were associated with a putative resistance to the disease. Both the haplotypes and genotypes were inferred by PHASE v.2.1 software. To the best of our knowledge, this type of investigation was conducted for the first time on pig livestock distributed in different regions of Central Italy. Thus, the obtained findings may be considered very important since they add useful information about swine genetic background in relation to PRRS infection, from the perspective of adopting Marker-Assisted Selection (MAS) as a possible and alternative strategy to control this still widespread disease.

## 1. Introduction

Porcine reproductive and respiratory syndrome (PRRS) was described in 1987 in the USA as a mysterious disease that affects swine, causing clinical signs like fever, weight loss, reproductive failure during late gestation, and respiratory distress [1]. Pigs are the only known hosts of the PRRS virus (PRRSV), characterized by a strict infection tropism for porcine monocytes/macrophages lineages [2]. To date, much knowledge about the disease and its etiological agent has been accumulated. For more than three decades, the disease, although not having zoonotic potential, has become one of the most widespread and impactful for the management of the pig industry mainly due to the high economic losses but also due to animal welfare issues. The causative agent is an enveloped positive single-stranded RNA virus from the family Arteviridae [3]. According to the current taxonomy, PRRSV exists in two distinct species, *Betaartevirus suid 1* (PRRSV-1), also known as the European (EU) strain, and *Betaartevirus suid 2* (PRRSV-2), recognized as the North American (NA) strain that share about 70% of their nucleotide sequences [4]. The host cells–virus interaction is quite complex, and this makes pathogen control still unsatisfactory, also because PRSSV has developed different strategies over time to counter the host’s antiviral response, providing favorable conditions for its survival [5]. In addition, the criticisms of disease management in animal husbandry mainly depend on the high genetic and antigenic variability of the virus and on the ability to result in prolonged infection in some individuals with subsequent virus shedding over long periods of time. PRRSV is constantly evolving, and the various circulating virus strains share only partial cross protection. Different type of vaccines are commercially available and effective in protecting against infection with highly homologous viral strains, but they provide incomplete protection toward heterologous strains. Moreover, the vaccine virus retains a certain degree of virulence, and it can be excreted by vaccinated subjects, infecting other susceptible animals, and thus its administration could have serious side effects. For these reasons, the role and the choice of vaccination against PRRSV is highly controversial [6]. With regard to the PRRSV genome, it is approximately 15 kb in length and is organized into 11 Open Reading Frames (ORFs) [7,8].

Concerning the host, different genes are involved in PRRSV recognition and infection, but this investigation focuses on the *CD163* gene since different studies have demonstrated, by now, its central role in viral susceptibility [9].

Porcine CD163 is a type I transmembrane protein that belongs to the class B of the scavenger receptor cysteine rich (SRCR) superfamily (SRCR-SF). It is one of the innate immunity receptors (PRRs—Pattern Recognition Receptors). It consists of 17 exons coding for a signal peptide, nine extracellular SRCR domain, two proline-serine-threonine (PST) domains, a single trans-membrane segment, and one intracellular domain. It is expressed in monocytes’ lineage and subpopulations of mature tissue macrophages [10,11,12]. CD163-positive macrophages, particularly porcine alveolar macrophages (PAMs), are the PRRSV primary target cells in vivo [7]. The main function of CD163 is the elimination of free hemoglobin (Hb) from the blood by binding the hemoglobin/haptoglobin complex (Hg/Hp) by the SCRC3 domain [9,10,12]. Moreover, CD163 has been recognized as an essential fusion receptor for PRRSV, and the SRCR5 domain has been shown to be an interaction site for the virus in vitro [13]. Indeed, the transfection of nonpermissive cells with CD163 cDNA was sufficient to render the cells fully permissive to the virus [14]. This was confirmed by gene knockout experiments in which the CD163 was deleted. These animals became resistant to the PRRSV-1 and PRRSV-2 isolates [15,16,17,18]. Recently, other studies confirmed that a partial or complete deletion by CRISPR/Cas 9 technology of the *CD163* gene exon 7, encoding SRCR5 domain, makes animals resistant to PRRSV infection without interfering with other biological functions [19]. In other controlled and conditioned trials and in experimental infections, some single nucleotide polymorphisms markers (SNPs) on different exons of the *CD163* gene were investigated to elucidate those related to susceptibility, resistance, or “resilience” profiles in relation to PRRSV infection [20,21,22]. Resilience to a disease is defined in terms of resistance and tolerance. The resistance is the ability of the host to resist infection when exposed to a pathogen, while the tolerance is the ability of an infected host to limit the damage caused by a pathogen [23]. Particularly in recent decades, veterinary research has aimed to identify alternative methods of preventing diseases with a non-chemotherapeutic approach that avoids, as far as possible, the exclusive and massive use of drugs [24]. Among these, the possibility of exploiting animal biological resources, genetic traits, and immune-genetic markers related to food quality, to animal welfare, and to a deeper resistance to pathogens of different nature is of great interest. Furthermore, disease resistance or susceptibility can depend also on the animal breeds. For example, for PRRS, it was demonstrated, by in vitro macrophage infection, that Landrace and Large White seem to be more susceptible than native and indigenous Chinese breeds like Tongcheng, Dapulian, or Jiangquhai, who show mild signs of the disease [25].

On this basis, the strategic goal of this study was to characterize some populations of commercial hybrids pig livestock located in different regions of Central Italy, in order to define animals potentially resistant/“resilient” to PRRSV infection. To the best of our knowledge, it is the first survey conducted in this field in pig populations that are naturally PRRSV exposed and/or infected and reared in these geographical areas. Specifically, we aimed to evaluate some polymorphic markers of the *CD163* gene, informative about the degree of response to the infection, in order to define the genotypes existing in the population object of the study. By comparing our data with those already present in the current literature and based on the association with the individual virological status, we tried to categorize animals in potentially susceptible and resistant/“resilient” groups. In the future, the most significant and robust genetic polymorphic markers related to PRRSV infection identified in this preliminary study and to be confirmed in other in depth investigations might be applied by the breeders and by the other stakeholders in genetic selection plans for pig breeds (MAS, Marker-Assisted Selection). This approach could be an important strategy for obtaining “PRRS-resistant” genetic lines, thus overcoming vaccination and disease management issues.

## 2. Materials and Methods

### 2.1. Investigated Pig Populations

For this study, 377 tissue samples (lung and diaphragm) of commercial hybrid pigs were collected. The animals, reared in different farms located in Central Italy, in particular, in the Umbria, Toscana, and Lazio regions, included two production types, fattening and/or finishing. Some of the tested pigs came from herds from Denmark and Netherlands that were certified PRRSV-free, and others came from Italian production chains declared PRRSV-free. The fattening and finishing steps were performed in the abovementioned Italian farms where vaccination for Aujeszky’s disease, which is compulsory by law in Italy, was practiced.

### 2.2. Samples Collection and RNA Extraction

Tissue aliquots were taken by authorized veterinarians after slaughter during the mandatory controls planned for the pig breeding sector; therefore, no ethical approval was required. Total RNA extraction was performed from the collected tissues for the genetic analysis of the host and from meat juice for the detection of viral RNA. In order to collect the meat juice, the tissue samples were initially frozen at −20 °C. At the time of analysis, the samples were allowed to thaw and were centrifuged at 4000 rpm for 10 min. Total RNA extraction was performed from about 80 mg of each collected tissue homogenized in 1 mL of Trifast (VWR company^®^, Avantor, Radnor Township, PA, USA), by TissueLyser II (Qiagen^®^, Hilden, Germany), with two steps at 20.0 Hz for 3′. After the homogenization phase, the RNA was purified, precipitated, and washed following the manufacturer’s instructions for the Trifast reagent (VWR company^®^, Avantor). The extracted RNA was resuspended in 50 μL of nuclease free water. Similarly, RNA was extracted from 250 µL of meat juice in 750 µL of Trifast^®^, with the same protocol used for the tissues described above, without the homogenization step. The RNAs’ quantity and quality were estimated photometrically with a Biophotometer (Eppendorf^®^, Hamburg, Germany).

### 2.3. CD163 RT-PCR and Sequencing 

Reverse Transcriptase (RT-PCR) assay optimization for the *CD163* target gene was performed, using the SuperscriptTM IV One Step RT-PCR System (Thermo Fisher Scientific, Waltham, MA, USA) and a primer set selected from Lim et al., 2017 [21], whose sequences are reported in Table 1. Amplification reactions were set up on a Mastercycler Ep Gradient S (Eppendorf^®^), and the oligonucleotides were purchased from Invitrogen, Thermo Fisher Scientific. The best PCR amplification conditions found, used for the analysis of the collected samples, are described as follows: a final volume of 50 µL containing about 650 ng of target RNA and 400 nM of the forward and reverse primers. The PCR protocol was carried out with the following thermal cycling profile: a retro-transcription step at 55 °C for 10 min, followed by an initial step of denaturation at 98 °C for 2 min, 35 cycles at 98 °C for 10 s, 62 °C for 10 s, 72 °C for 40 s, and a final extension step at 72 °C for 5 min. 

*CD163* amplification was controlled on 1.5% agarose gel electrophoresis containing Midori Green Advanced DNA Stain (Nippon Genetics Europe GmbH, Düren, Germany). The PCR products were purified with the QIAquick^®^ PCR Purification Kit (Qiagen), according to the manufacturer’s instructions, and eluted in a final volume of 30 µL. The quantity and quality of the PCR products were assessed photometrically with a Biophotometer (Eppendorf^®^). The sequencing reactions were performed using the BrilliantDyeTM Terminator Cycle Sequencing Kit v3.1 (NimaGen BV, Nijmegen, The Netherlands) in accordance with the manufacturer’s instructions. Sequencing reactions were run in a 3500 Genetic Analyzer (Thermo Fisher Scientific). All sequences, in FASTA format, were aligned to *Sus scrofa CD163* mRNA complete cds (GenBank Accession Number: DQ067278.1) and analyzed with BioEdit v7.2.5 software [26], using the ClustalW algorithm. 

### 2.4. PRRSV RT-qPCR

Viral RNAs, extracted from meat juice, were subjected to qualitative RT-qPCR, using VetMAX™ PRRSV EU & NA 2.0 kit (Applied Biosystems, Foster City, CA, USA), targeting the PRRSV-1 (EU) and PRRSV-2 (NA) strains, following the manufacturer’s instructions. The samples were tested in duplicate.

RT-qPCR amplification was performed using a QuantStudio™ 7 Flex Real-Time PCR System (Thermo Fisher Scientific) using the following thermal cycling conditions: a retro-transcription step at 50 °C for 5 min, an initial step at 95 °C for 10 min, followed by 40 cycles at 95 °C for 3 s and 60 °C for 30 s. The results were analyzed using the QuantStudio™ 7 Flex Software (Thermo Fisher Scientific). Positive and negative controls were introduced in each analytical session. In particular, negative controls for the extraction and amplification steps were included, and, in addition, an external positive control (EPC) and an internal positive control (IPC) provided by the abovementioned kit (Applied Biosystems) were used, following the manufacturer’s instructions. Positive and negative results were assigned to the analyzed samples as indicated by the kit datasheet: positive samples (Cq < 40) and negative samples (Cq > 40).

### 2.5. Statistical Analysis and Data Elaboration

The most probable haplotypes and genotypes of the *CD163* gene were calculated using PHASE v2.1 software [27] that performs the best reconstruction of haplotypes/genotypes in a population, based on Bayesian inference, so on probabilistic events. The analysis of pairwise linkage disequilibrium (LD) among SNPs was performed by means of the SHEsis Plus software (https://www.snpstats.net/) [28,29,30], and the LD extent has been expressed by means of the D’ coefficient. The SNP genotypes of infected and healthy pigs were compared, and the risk associated with the genotypes was estimated as an odds ratio (OR) with 95% confidence intervals (95% CI). The most frequent homozygous genotype, observed in the studied population, was set as the baseline. The statistical analysis was performed by SNPStats software (https://pubmed.ncbi.nlm.nih.gov/19290020/) [31]. The genotypes of the animals were compared, and the ORs, with a 95% confidence interval, were calculated, setting the most frequent genotype as the baseline. The ORs were calculated according to Altman (1991) [32], and Haldane’s correction was applied for ORs that involved the genotypes 3/16, 9/16, 2/30, and 16/19 [33]. A *p*-value < 0.05 was considered statistically significant for all the analyses. 

## 3. Results

### 3.1. Association between SNPs and Risk of PRRS Infection 

From the *CD163* gene sequences analysis, 11 polymorphic sites were identified. Information about the investigated polymorphisms is reported in Table 2. The allelic frequencies of the analyzed SNPs, obtained in the studied animal population, are reported in Table 3.

In order to evaluate the potential effect of the *CD163* gene polymorphisms on disease resistance or susceptibility, based on the results obtained by the PRRSV RT-qPCR assay, we divided the pigs in two groups: infected animals (RT-qPCR positive) and healthy animals (RT-qPCR negative). The linkage disequilibrium D’ coefficients between all the SNPs pairs are shown in Appendix A.

The risk of the SNPs genotypes to become sick was calculated using a logistic regression analysis. The results showed that three SNPs were significantly associated with a low risk of PRRS infection. Particularly, pigs with heterozygous genotypes at the G2509C, G2638A, and C3534T sites had a low relative risk (OR = 0.90, *p* = 0.005; OR = 0.38, *p* = 0.022; OR = 0.43, *p* = 0.0039, respectively). Another heterozygous genotype at C3082T polymorphic site seemed to be protective with an OR = 0.46 but with a *p* value of 0.052, which was not a significant value but very close to the significance threshold. On the contrary, the homozygous C/C genotype at G2509C and the T/T genotype at C3534T had a greater relative risk (OR = 3.11, *p* = 0.0054 and OR = 1.91, *p* = 0.0039), as shown in Table 4.

### 3.2. Association between the Animal Genotypes and the Risk of PRRS Infection

The 11 SNPs found polymorphic in the studied population gave rise to 36 haplotypes, whose combination provided 62 genotypes. The haplotypes reconstructed by the PHASE v2.1 software, were numbered with progressive numbering (1–36), and the genotypes were indicated by the two numbers of the haplotypes that built them up (i.e., 16/30). Out of the 62 total genotypes, also in this case identified by the PHASE v2.1 software, only the 10 genotypes found with frequencies > 1% (16/16; 16/30; 3/9; 3/16; 9/30; 3/30; 30/30; 9/16; 2/30; 16/19) were considered and included in the statistical analysis. The frequencies of the above mentioned 10 genotypes ranged from 1.06% to 17.2% (Appendix A). The analysis of the animal genotypes showed that three genotypes (16/30, 3/9, and 3/16) had a significantly very low relative risk of infection (OR = 0.351, *p* = 0.0442; OR = 0.072, *p* = 0.0130 and OR = 0.042, *p* = 0.0300, respectively), while the other genotypes did not have significant *p*-values (Table 5).

## 4. Discussion

This investigation was conducted *post-mortem* in PRRSV naturally exposed and/or infected pig livestock. The animals were reared in farms most of which were of relatively recent construction (late 1990s–early 2000s) and subsequently restructured according to the recent European regulations on animal welfare, in particular concerning biosecurity measures and environmental control. On this basis, we can assume that the environmental effects were limited or irrelevant for the study purposes. 

By now, several studies have shown that the SRCR domain of the CD163 receptor plays a key role in the internalization of the PRRSV into lung macrophages at the onset of infection [8]. These findings are supported by the fact that by transfecting cells nonpermissive to the virus with *CD163* cDNA, the cells become fully permissive [14]. Some natural mutations have been described in this domain, and they can be considered either as protective factors when they are associated with a better response of the animal to the virus or as risk factors when they are associated with a greater probability of infection. For example, the CC genotype of *CD163* at the C3534T polymorphism has been found to be significantly associated with low IgG levels after the PRRS challenge [21], while, on the other hand, the AA genotype of G2494A polymorphism had a significant association with susceptibility to infection [34]. To date, most of the investigations focused on the evaluation of the gene polymorphic variants’ effect by means of controlled and conditioned trials or experimental infections [20,21,22]. On the other hand, our study focused on a preliminary association between single significant polymorphisms in the *CD163* target gene found in the controlled population and the relative risk of PRRS infection. The main aim was the assessment of the association between the specific SNPs’ genotypes with viral presence, thus with host infection status, in naturally exposed and/or infected pigs. In particular, in order to correctly categorize the animals into two different phenotypes to be associated with genetic profiles, the study was conducted on swine whose virological status was determined by the gold standard technique, namely RT-qPCR. 

In order to add useful information about the association between the PRRS virological status and resistance/resilience genetic profiles in pig livestock, we started from the literature data and, in particular, from the functional and challenge experiments conducted by Dong et al., 2021 [22]. These authors performed an in depth analysis also evaluating the *CD163* gene SNPs, located on different exons (9–15), and their association with PRRSV infection status in different conditions. We then paid attention to the *CD163* regions, encompassing exons 11–15, where the candidate SNPs had a more significant role in the response to PRRS infection. In particular, among 11 polymorphic SNPs detected, we found four SNPs, (G2509C, G2638A, C3082C, and C3534T), which were significantly associated with a low risk of infection if in heterozygous form. These results are in agreement with Dong et al., 2021 [22] who found that some heterozygous SNPs in the *CD163* gene had a more favorable host response towards PRRS infection. However, while Ren et al., 2012 [20] found in the Yichang and Xiangfan native Chinese pig breed that the AA of *CD163* at the polymorphic site A2592G seemed to be associated with PRRSV infection resistance, in our population, the A/A genotype, although present in 183 healthy pigs (54.3%), had a OR value equal to 1, so it did not exhibit a risk or protective role.

Regarding the animals’ genotypes, among the most frequent ones, three had a significantly protective effect (16,30; 3,9; 3,16 genotypes). On this basis, we can therefore hypothesize that some *CD163* mutant genotypes may exert a protective role against PRRSV infection. As alternative to traditional approach in disease control, different studies in vitro were conducted to understand the mechanisms helpful to contrast the PRRS infection, such as the treatment of cell lines with bioactive compounds from *Caesalpinia sappan* extract, that seemed to inhibit the PRRSV-CD163 interaction, successfully limiting the number of PRRSV RNA copies [35]. Furthermore, the application of genetic and molecular tools could be another possibility for the control of PRRSV infections. Indeed, the implementation of MAS schemes is a strategy that allows the increase in resistance/resilience traits to infectious diseases while ensuring animal welfare and the reduction in farm management costs also due to the decrease in pharmacological treatments, with the related public health risks, to be undertaken on affected farms. Thus, the economic value of resilience increases as the number of animals rises and can become as large as the economic value of production. So, resilience should be built into breeding programs. Finally, our study was restricted to a portion of the *CD163* gene, whose observed polymorphic variants were assessed by the Sanger approach. Thus, further and in-depth investigations, also based on high throughput next generation sequencing, which allows multitarget and multivariant analyses, could be conducted on larger numbers of animals, coming also from other geographical areas, in order to confirm and then eventually apply these promising findings.

## 5. Conclusions

Without effective vaccines, different and alternative approaches to fight infectious diseases are needed. The exploitation of the host’s genetic resources should be added to other strategies already implemented such as biosecurity measures. To date, few studies have been conducted to identify protective genetic variants in naturally PRRSV exposed and/or infected pig populations. In this work, we suggest that selecting swine for some polymorphic markers, such as G2509C, G2638A, C3082C and C3534T, is expected to increase the animals’ response to PRRSV. Significant associations of the *CD163* variants with resilience/susceptibility were found in the investigated pig population; in particular, the combination of the abovementioned SNPs lead to genotypes that appear to be protective factors for the host against PRRSV infection. To confirm the preliminary results of our work, further investigations would be necessary, with the ultimate goal of adopting and implementing MAS programs, by crossing animals carrying these protective variants. Indeed, the exclusion of susceptible animals predisposed to PRRS would limit both the development of clinical symptoms and the spread of the virus, also reducing the economic losses resulting from the disease and the public health risks linked to vaccination and mainly to drug use. In conclusion, a genetics-based approach could be an innovative strategy to control PRRSV infection, especially in high-productivity pig herds. 

## Figures and Tables

**Table 1 animals-13-02477-t001:** Primer pair sequences of the *CD163* target gene (cDNA) investigated in the study.

Target Gene	Primer Sequence 5′--->3′	Amplicon Length	Reference
*CD163*	For TTAATGCCACTGGTTCTGCTC	1411 bp	Lim et al., 2017 [21]
	Rev TGCCCTTGAAAGTCTTACATA		

**Table 2 animals-13-02477-t002:** Information on the selected and investigated SNPs in the *CD163* target gene.

SNP	SNP ID	Chr	Location 1	ID Variant	Allele 1	Allele 2	Location	Type of Mutation
c.2494G>A	CD163_SNP1	5	63,325,006	rs1107556229	G	A	Exon 10	synonymous
c.2509G>C	CD163_SNP2	5	63,326,686	/	G	C	Exon 11	synonymous
c.2592A>G	CD163_SNP3	5	63,326,769	/	A	G	Exon 11	missense
c.2638G>A	CD163_SNP4	5	63,326,815	/	G	A	Exon 11	synonymous
c.2935G>A	CD163_SNP5	5	63,327,893	rs81215636	G	A	Exon 12	synonymous
c.2983C>A	CD163_SNP6	5	63,327,941	rs81215637	C	A	Exon 12	synonymous
c.3082C>T	CD163_SNP7	5	63,328,040	rs81215638	C	T	Exon 12	synonymous
c.3121T>C	CD163_SNP8	5	63,328,079	rs1111118836	T	C	Exon 12	synonymous
c.3346G>A	CD163_SNP9	5	63,330,243	/	G	A	Exon 14	3′ UTR
c.3534C>T	CD163_SNP10	5	63,334,407	/	C	T	Exon 15	3′ UTR
c.3547A>G	CD163_SNP11	5	63,334,420	/	A	G	Exon 15	3′ UTR

SNPs are annotated based on build 11.1 of the pig genome (Ensembl), Allele 1 indicates the major allele (more frequent), Chr: chromosome; ID variant: coordinate of polymorphism location at chromosome locus). UTR: untranslated region.

**Table 3 animals-13-02477-t003:** Allelic frequencies of 11 SNPs in the *CD163* gene.

		All Subjects		Healthy		Infected	
SNP	Allele	Count	Frequency *	Count	Frequency *	Count	Frequency *
G2494A							
	G	564	0.75	506	0.75	58	0.72
	A	190	0.25	168	0.25	22	0.28
G2509C							
	G	461	0.61	424	0.63	37	0.46
	C	293	0.39	250	0.37	43	0.54
A2592G							
	A	558	0.74	499	0.74	59	0.74
	G	196	0.26	175	0.26	21	0.26
G2638A							
	G	642	0.85	568	0.84	74	0.92
	A	112	0.15	106	0.16	6	0.08
G2935A							
	G	749	0.99	671	1.00	78	0.98
	A	5	0.01	3	0.00	2	0.02
C2983A							
	C	749	0.99	671	1.00	78	0.98
	A	5	0.01	3	0.00	2	0.02
C3082T							
	C	640	0.85	567	0.84	73	0.91
	T	114	0.15	107	0.16	7	0.09
C3121T							
	T	535	0.71	481	0.71	54	0.68
	C	219	0.29	193	0.29	26	0.32
G3346A							
	G	627	0.83	555	0.82	72	0.9
	A	127	0.17	119	0.18	8	0.1
C3534T							
	C	450	0.6	408	0.61	42	0.52
	T	304	0.4	266	0.39	38	0.48
G3547A							
	A	545	0.72	488	0.72	57	0.71
	G	209	0.28	186	0.28	23	0.29

* Frequencies’ values were rounded to the second decimal figure.

**Table 4 animals-13-02477-t004:** Genotypic frequencies, odds ratio (OR), and association with the risk of pigs to be PRRSV infected.

SNP Genotype	Healthy Pigs	Infected Pigs	OR (95% CI)	*p*-Value
G2494A				
G/G	190 (56.4%)	20 (50%)	1.00	0.64
G/A	126 (37.4%)	18 (45%)	1.36 (0.69–2.67)	
A/A	21 (6.2%)	2 (5%)	0.90 (0.20–4.14)	
G2509C				
G/G	144 (42.7%)	13 (32.5%)	1.00	0.0054
G/C	136 (40.4%)	11 (27.5%)	0.90 (0.39–2.07)	
C/C	57 (16.9%)	16 (40%)	3.11 (1.41–6.88)	
A2592G				
A/A	183 (54.3%)	20 (50%)	1.00	0.42
A/G	133 (39.5%)	19 (47.5%)	1.31 (0.67–2.55)	
G/G	21 (6.2%)	1 (2.5%)	0.44 (0.06–3.41)	
G2638A				
G/G	231 (68.5%)	34 (85%)	1.00	0.022
G/A	106 (31.4%)	6 (15%)	0.38 (0.16–0.94)	
G2935A				
G/G	334 (99.1%)	38 (95%)	1.00	0.084
G/A	3 (0.9%)	2 (5%)	5.86 (0.95–36.18)	
C2983A				
C/C	334 (99.1%)	38 (95%)	1.00	0.084
C/A	3 (0.9%)	2 (5%)	5.86 (0.95–36.18)	
C3082T				
C/C	230 (68.2%)	33 (82.5%)	1.00	0.052 *
C/T	107 (31.8%)	7 (17.5%)	0.46 (0.20–1.06)	
C3121T				
T/T	170 (50.5%)	17 (42.5%)	1.00	0.6
C/T	141 (41.8%)	20 (50%)	1.42 (0.72–2.81)	
C/C	26 (7.7%)	3 (7.5%)	1.15 (0.32–4.21)	
G3346A				
G/G	221 (65.6%)	32 (80%)	1.00	0.13
G/A	113 (33.5%)	8 (20%)	0.49 (0.22–1.10)	
A/A	3 (0.9%)	0 (0%)	0.00 (0.00–NA)	
C3534T				
C/C	132 (39.2%)	17 (42.5%)	1.00	0.0039
C/T	144 (42.7%)	8 (20%)	0.43 (0.18–1.03)	
T/T	61 (18.1%)	15 (37.5%)	1.91 (0.89–4.07)	
G3547A				
A/A	175 (51.9%)	19 (47.5%)	1.00	0.69
G/A	138 (41%)	19 (47.5%)	1.27 (0.65–2.49)	
*G/G*	24 (7.1%)	2 (5%)	0.77 (0.17–3.50)	

* Value close to the significant threshold; OR: Odds Ratio; 95% CI: 95% Confidence Interval.

**Table 5 animals-13-02477-t005:** Pigs’ genotypes, Odds Ratio (OR), and association with the risk of infection.

Genotype	Infected Pigs (N)	Healthy Pigs (N)	OR	95% CI	*p*-Value **
(16,30)	6	57	0.351	0.127–0.973	0.0442
(3,9)	1	46	0.072	0.009–0.571	0.0130
(3,16)	0	38	0.042 *	0.002–0.729	0.0300
(9,30)	3	29	0.345	0.092–1.293	0.1143
(3,30)	6	14	1.429	0.468–4.365	0.5314
(30,30)	1	18	0.185	0.023–1.504	0.1146
(9,16)	0	16	0.099 *	0.006–1.742	0.1139
(2,30)	0	6	0.251 *	0.013–4.704	0.355
(16,19)	0	4	0.362 *	0.018–7.104	0.5035

Each genotype is compared to the most frequent genotype (16,16); OR: Odds Ratio; 95% CI: 95% confidence interval; numbers in brackets: genotypes derived from the combination of the two haplotypes whose numbers were assigned by PHASE v2.1 analysis; * Haldane correction; ** *p*-values not corrected for multiple-hypothesis testing.

## Data Availability

Data supporting the reported results are available in this article and in the Appendix A.

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
