# Peer review of "Porcine Reproductive and Respiratory Syndrome (PRRS) and CD163 Resistance Polymorphic Markers: What Is the Scenario in Naturally Infected Pig Livestock in Central Italy?"

_animals, 2023, doi:10.3390/ani13152477_

Round 1
Reviewer 1 Report
In this manuscirpt, the authors found that pigs with heterozygous genotypes at G2509C, G2638A, and C3534T sites in CD163, had a low relative risk of PRRS in the Italian pig population. This study is very interesting and the candidate SNPs may be important for breeding PRRSV resistant pigs. However, the biggest problem is that candidate SNPs have not been validated against PRRSV infection at the cellular level by base-editing techniques.
1.In Table 5, Genotype description is unclear; What is OR and CI?
2.For the result of “Association between animals genotypes and risk of PRRS infection”, the methodology and conclusions may be incorrect. It may not be correct to detect infection or non-infection in pigs by PRRSV alone. Since not actively infected with PRRSV, it is not clear if the pigs have been infected or treated and cured by medication? It is not clear whether the pigs are infected with PRRSV only, as they are not SPF grade animal houses.
3.Although some polymorphic markers, such as G2509C, G2638A, C3082C and C3534T was expected to increase the animals' response to PRRSV. The author believes that these SNPs lead to genotypes that appear to be protective factors for the host against the infection. There is a lack of data to support whether these candidate SNPs are associated with PRRSV infection due to the lack of functional experiments.
The quality of English language is good.
Author Response
Reviewer 1
Inizio modulo
Open Review
( ) I would not like to sign my review report
(x) I would like to sign my review report
Quality of English Language
( ) I am not qualified to assess the quality of English in this paper
( ) English very difficult to understand/incomprehensible
( ) Extensive editing of English language required
(x) Moderate editing of English language required
( ) Minor editing of English language required
( ) English language fine. No issues detected
|
Yes |
Can be improved |
Must be improved |
Not applicable |
|
|
Does the introduction provide sufficient background and include all relevant references? |
( ) |
(x) |
( ) |
( ) |
|
Are all the cited references relevant to the research? |
( ) |
(x) |
( ) |
( ) |
|
Is the research design appropriate? |
( ) |
( ) |
( ) |
(x) |
|
Are the methods adequately described? |
( ) |
( ) |
(x) |
( ) |
|
Are the results clearly presented? |
( ) |
( ) |
(x) |
( ) |
|
Are the conclusions supported by the results? |
( ) |
( ) |
( ) |
(x) |
Comments and Suggestions for Authors
Response to Reviewer #1
In this manuscript, the authors found that pigs with heterozygous genotypes at G2509C, G2638A, and C3534T sites in CD163, had a low relative risk of PRRS in the Italian pig population. This study is very interesting and the candidate SNPs may be important for breeding PRRSV resistant pigs. However, the biggest problem is that candidate SNPs have not been validated against PRRSV infection at the cellular level by base-editing techniques.
We are pleased that our manuscript is considered interesting and valuable.
With regards to the last observation, we want to specify that the trials and validation against PRRSV infection at the cellular level by base-editing techniques, already conducted in other studies (i.e Chen et al., 2019; Cheng et al., 2020), were not the current aims of the work. The goal of our researh was to evaluate polymorphisms, naturally present in the host, and their effect on response to PRRS infection in real condition of breeding. We think that our results could be useful to conduct further in depth analyses, also in the direction you suggested, on a larger number of animals, prior to the exploitation of genetic resources to be applied for Marker Assisted Selection, based on crossing the animals carriers of resistant/”resilient” genetic traits/haplotypes.
1.In Table 5, Genotype description is unclear; What is OR and CI?
Thanks for the comment. We clarified the captions about genotype description (Table 5) and we explained OR and CI acronyms reported both in Table 5 and in Table 4.
- For the result of “Association between animals genotypes and risk of PRRS infection”, the methodology and conclusions may be incorrect. It may not be correct to detect infection or non-infection in pigs by PRRSV alone. Since not actively infected with PRRSV, it is not clear if the pigs have been infected or treated and cured by medication? It is not clear whether the pigs are infected with PRRSV only, as they are not SPF grade animal houses.
In this study, phenotype categorization was based on viral detection in diaphragmatic and lung meat juices by RT-qPCR, that is currently the most widely used test. These tests are rapid, highly sensitive, specific and can detect the presence of PRRSV in various matrices. RT-qPCR is the gold standard and the technique of choice for diagnosing disease in breeding and growing animals also for monitoring the success of disease control programmes (Holtkamp et al., 2011).
Unfortunately, we had not reported before this aspect in the previously submitted manuscript, but actually some tested pigs came from Denmark and Netherlands herds that were certified PRRSV-free (by means of certification’s release from the countries of origin) and other came from Italian production chains declared PRRSV-free. Anyway, the fattening and finishing steps, for all the pigs, were performed in the Italian farms in object where vaccination for Aujeszky's disease, which is compulsory by law in Italy, was practiced. This clarification was partially inserted in the main text in particular in “Material and methods” section, 2.1. “Investigated pig populations” at lines 145-149.
Furthermore, the examined pigs came from in-door farms, where the health status of animals was under control. Indeed, most of the investigated farms are of relatively recent construction (late 1990s-early 2000s) and subsequently restructured according to the recent European regulations on animal welfare, in particular concerning the biosecurity measures and environmental control. This aspect was clarified in the main text, in particular in “Discussion” section, at lines 275-280.
Anyway, the SPF condition can not be ensured and, in this study, the co-morbidity with other pathogens was not assessed.
- Although some polymorphic markers, such as G2509C, G2638A, C3082C and C3534T was expected to increase the animals' response to PRRSV. The author believes that these SNPs lead to genotypes that appear to be protective factors for the host against the infection. There is a lack of data to support whether these candidate SNPs are associated with PRRSV infection due to the lack of functional experiments.
Our work aimed to be a preliminary investigation conducted for the first time in finishing/fattening farms located in some regions of Central Italy. The primary scope was to add useful information about the association between PRRS virological status and resistance/resilience genetic profiles in pig livestock naturally exposed to the virus. In order to reach this goal, we started from the literature data and, in particular, from the functional and challenge experiments conducted by Dong et al. 2021. These authors performed an in depth analysis also evaluating CD163 gene SNPs, located on different exons (9-15), and their association with PRRSV infection status, in different conditions. We then paid attention on the regions (exons 11-15) where the candidate SNPs had a higher significance in the PRRS infection. Furthermore, our data are in accordance with those obtained by Dong et al 2021. We added this clarification in the main text of the manuscript in “Discussion” Section at lines 304-308.
Comments on the Quality of English Language
The quality of English language is good.
Thanks for your evaluation about English language.

Reviewer 2 Report
The study investigates host resistance to Porcine Reproductive and Respiratory Syndrome (PRRS) virus in slaughtered pigs from Central Italy.
The general part of the introduction may be shortened to leave more room for the own results. Text in lines 33-38 should be shortened.
Here the question may arise how did you regard environmental stressors in your study design. This question is addressed but not resolved in the Introduction.
This report in SciRep 2022 Dec 14;12(1):21595. doi: 10.1038/s41598-022-26206-x may be also relevant how to prevent PRRSV infection in pigs.
There were proposed other genes playing a role in host resistance/resilience. See the review in Virus Res 2023 Mar; 326: 199057. A supplementary Table would be useful to summarize genes and variants already known. Breeds less susceptible to PRRS infection may be also mentioned, e.g., Jiangquhai pig.
Table 1. Lenght >> Length
Line 153: CD163 RT-PCR and sequencing. Can you please outline which variants you expect to find in the amplicon and which role these have for the host.
PRRSV RT-qPCR: positive and negative controls: how many did you use. How did you define the reference curve. How did you define infected and non-infected animals. Please give your reference value.
Table 5. point instead of comma. Please also account for multiple comparisons.
Line 257: polimorphism >> polymorphism
Line 259: suscetiptibility >>> susceptibility
Discussion focuses on CD163 variants but does not explain the limitations of the approch used. It is not clear which other variants may influence infection status. Furthermore, environmental effects are not considered in the discussion.
In order to get a clearer picture on the use of genotyping CD163 variants, the discussion has to include the possible role of other variants and to explain the limitations of the present study.
THe authors should also explain which approach may be be more informative to capture a larger spectrum of variants involve in host resilience and resistance.
Sci Rep
. 2022 Dec 14;12(1):21595. doi: 10.1038/s41598-022-26206-x. Sci Rep
. 2022 Dec 14;12(1):21595. doi: 10.1038/s41598-022-26206-x.
No comment
Author Response
Reviewer 2
Open Review
( ) I would not like to sign my review report
(x) I would like to sign my review report
Quality of English Language
( ) I am not qualified to assess the quality of English in this paper
( ) English very difficult to understand/incomprehensible
( ) Extensive editing of English language required
(x) Moderate editing of English language required
( ) Minor editing of English language required
( ) English language fine. No issues detected
|
Yes |
Can be improved |
Must be improved |
Not applicable |
|
|
Does the introduction provide sufficient background and include all relevant references? |
( ) |
(x) |
( ) |
( ) |
|
Are all the cited references relevant to the research? |
(x) |
( ) |
( ) |
( ) |
|
Is the research design appropriate? |
(x) |
( ) |
( ) |
( ) |
|
Are the methods adequately described? |
( ) |
(x) |
( ) |
( ) |
|
Are the results clearly presented? |
( ) |
(x) |
( ) |
( ) |
|
Are the conclusions supported by the results? |
( ) |
(x) |
( ) |
( ) |
Comments and Suggestions for Authors
Response to Reviewer #2
The study investigates host resistance to Porcine Reproductive and Respiratory Syndrome (PRRS) virus in slaughtered pigs from Central Italy.
The general part of the introduction may be shortened to leave more room for the own results. Text in lines 33-38 should be shortened.
In the Abstract we removed a redundant sentence, at lines 34-35, present also in the main text, as suggested.
As indicated by the reviewer, we tried also to shorten as much as possible the “Introduction” section.
Here the question may arise how did you regard environmental stressors in your study design. This question is addressed but not resolved in the Introduction.
The mention of environmental stressors was reported in “Abstract” section, since environment is one of the contributing factors that influences conditioned infectious diseases (mainly the multifactorial and polygenic ones) together with genetic traits and epigenetic patterns.
Our research is not a controlled and standardised trial, but it is an investigation post-mortem conducted in PRRSV naturally exposed and/or infected pig livestocks; the assessment of environmental stressors (including i.e: co-morbidity, monitoring of breeding temperature and ventilation, feeding…) effect was not foreseen.
In particular, we have collected lungs and diaphragms of post-slaughter swine, deriving from different finishing/fattening pig farms located in Central Italy regions. Most of them are farms of relatively recent construction (late 1990s-early 2000s) and subsequently restructured according to the recent European regulations on animal welfare, in particular for biosecurity measures and environmental control. Furthermore, in the majority of cases the farms belong to large production chains and, even if located in different sites, all the breeding structures in object are built following the same criteria (i.e: biosecurity measures; monitoring of breeding temperature and ventilation by means of computer control systems; type of paving).
On this basis, we can assume that the stressors and environmental effects are limited or irrelevant for this study and for its purposes, consisting on a preliminary statistical association between CD163 host genotype and PRRS virological status qualitative evaluation (not on quantitative assessment of the viral load).
This aspect was clarified in the main text, in particular in “Discussion” section at lines 275-280.
This report in SciRep 2022 Dec 14;12(1):21595. doi: 10.1038/s41598-022-26206-x may be also relevant how to prevent PRRSV infection in pigs.
We added this reference as suggested by the reviewer, but in the “Discussion” section at lines 320-324, where in our opinion it was more pertinent.
There were proposed other genes playing a role in host resistance/resilience. See the review in Virus Res 2023 Mar; 326: 199057. A supplementary Table would be useful to summarize genes and variants already known. Breeds less susceptible to PRRS infection may be also mentioned, e.g., Jiangquhai pig.
Thanks for the comment. We are aware that different genes of the host may be involved in PRRSV infection. Our study focused on the CD163 gene, because, as reported in literature (i.e Reiner, 2016), it was identified as the most likely candidate involved in PRSSV infection, together with CD169 gene. However, by gene-editing techniques, particularly CRISPR‐Cas13a (i.e: Chen et al., 2019; Chang et al., 2020), only CD163 has been shown capable of conferring PRRSV (both EU and US strains/genotypes) permissiveness to cell lines unsusceptible to the virus, even in the absence of CD169 gene. In other words, CD163 is the receptor mostly studied, also for the genetic analyses, due to its key role in viral recognition, in particular at SRCR domain 5 region. We think that a Supplementary Table summerizing other genes and already known variants could be informative, but it could be misleading and not necessary for our study, that is not thought and designed as a “Review manuscript”. In the introduction, at lines 83-85, we indicated the reasons why we chose to investigate the CD163 gene.
Our study was conducted on commercial breeds and hybrids pigs. Anyway, as suggested, we specified in the main text, in particular in “Introduction” section at lines 115-119, that some breeds are more or less susceptible than others.
Table 1. Lenght >> Length
We have modified the term, accordingly.
Line 153: CD163 RT-PCR and sequencing. Can you please outline which variants you expect to find in the amplicon and which role these have for the host.
The polymorphic variants, detected by sequencing, were the same observed by Dong et al 2021 and de novo SNPs were not found in CD163 target sequence investigated in our study. According to the literature and as reported in “Conclusion” section of our manuscript the role of G2509C, G2638A, C3082C and C3534T in heterozygous form as well as the genotypes 16,30; 3,9; 3,16 are associated to a favorable host responses, so to a less susceptibility or low risk of PRRS infection.
PRRSV RT-qPCR: positive and negative controls: how many did you use. How did you define the reference curve. How did you define infected and non-infected animals. Please give your reference value.
Thanks for the observation. The required specifications were added in “Materials and Methods” section in the paragraph “2.4. PRRSV RT-qPCR”, at lines 201-206. Anyway, the PRRSV RT-qPCR is a qualitative assay, so reference curve was not necessary and it was not defined.
Table 5. point instead of comma. Please also account for multiple comparisons.
We inserted and replaced point instead of comma in Table 5, as you rightly suggested.
We did not account for multiple comparisons because the computation based on comparing the obtained genotypes with the most frequently genotype observed in our population (also in terms of OR calculation) was more pertinent for the study: it was established a priori and not post hoc. Anyway, for more clarity, we added in caption of Table 5 “p-values not corrected for multiple-hypothesis testing”.
Line 257: polimorphism >> polymorphism
Line 259: suscetiptibility >>> susceptibility
We modified the terms, accordingly.
Discussion focuses on CD163 variants but does not explain the limitations of the approch used. It is not clear which other variants may influence infection status. Furthermore, environmental effects are not considered in the discussion.
In order to get a clearer picture on the use of genotyping CD163 variants, the discussion has to include the possible role of other variants and to explain the limitations of the present study.
The authors should also explain which approach may be be more informative to capture a larger spectrum of variants involve in host resilience and resistance.
The aspects related to environmental effects have been already explained in response to your previous observations, and we have added the required clarification in the “Discussion” section at lines 275-280.
Discussion section was in general implemented and the limitations of this study as well as the future perspectives were added in Discussion section at lines 332-337.
Comments on the Quality of English Language
No comment

Round 2
Reviewer 2 Report
The authors regarded all comments and amended their manuscript accordingly.
No comments